# Thermal Creep Behavior and Creep Crystallization of Al-Mg-Si Aluminum Alloys

**DOI:** 10.3390/ma15228117

**Published:** 2022-11-16

**Authors:** Qinmin Zhang, Xiaomin Huang, Ran Guo, Dongyu Chen

**Affiliations:** Faculty of Civil Engineering and Mechanics, Kunming University of Science and Technology, Kunming 650500, China

**Keywords:** 6082 aluminum alloy, thermal creep, intrinsic model, creep crystallization, critical conditions

## Abstract

The experimental temperature is 613.15~763.15 K, and the strain rate is 0.01~10 s^−1^. The hot compression creep test of the 6082-T6 aluminum alloy sample is carried out by Gleeble-3500 hot compression simulation compressor, and its creep behavior is studied by scanning electron microscope. The results show that the DRX crystal has an irregular shape and that content of the Mg phase, Si phase, and Mn phase in the crystal are the main factors to change the color of DRX crystal. Temperature and strain rate are important factors affecting dynamic recrystallization. Reducing temperature and increasing strain rate will weaken dynamic recrystallization, and DRX critical condition and peak stress (strain) will increase. The constitutive equation of hot creep of 6082 aluminum alloy was established by introducing the work hardening rate-rheological stress curve, and the relationship between DRX critical condition, peak stress (strain) and parameter Z during creep was explored. Based on the Av rami equation, the prediction equation of the DRX volume fraction is established. With the increase of strain, DRX volume fraction is characterized by slow increase, then rapid increase and then slowly increase. In the hot -forming extrusion process of 6082 aluminum alloy, according to the volume fraction prediction equation, the DRX can be reduced, and the internal structure of the material can be optimized by changing the extrusion conditions and particle size.

## 1. Introduction

Al-Mg-Si alloy 6082 has outstanding all-around features, especially high strength, simple processing, and high corrosion resistance [1,2]. It has been extensively utilized in building, shipbuilding, fast trains, rail transportation, and automobiles [3]. In recent years, domestic and foreign scholars have carried out extensive research on the mechanical properties of aluminum alloys [4,5,6,7,8,9,10,11,12,13]. In the room temperature environment, the aluminum alloy is difficult to form and prone to rebound cracking, and the internal microstructure evolution is disturbed by more factors during the aluminum alloy thermoforming process, for example, deformation, loading rate, ambient temperature, and the key factor affecting the material properties is the change of microstructure. Softening phenomena such as DRV and DXR often occur in conjunction with the thermal deformation of the metal. For the 6082-aluminum alloy, current research is focused on corrosion, machining processes, and weldability [14,15,16,17,18,19]. Numerous scholars have also conducted studies on the heat deformation behavior of metals. Zhen et al. [20] conducted compound extrusion experiments (both direct and indirect extrusion) on solid solution-treated AA7050 aluminum alloy at 440 °C with large deformation. The microstructure of typical regions with different filler contents was qualitatively described and quantitatively characterized by the electron backscattered diffraction (EBSD) technique. The results showed that the recrystallization fractions of AF (ahead of filling), BF (beginning of filling), MF (mid-stage of filling) and EF during direct extrusion were 8.3%, 13.5%, 9.3% and 11.2%, respectively. The recrystallization fraction during indirect extrusion was 15.5%, 9.1%, 5.2% and 9.9%, respectively. This indicates that the mode and size of deformation play an important role in DRX. Hu et al. [21] studied the microstructure evolution of 7A85 aluminum alloy at strain rate (0.0011 S^−1^) and deformation temperature (250–450 °C) using optical microscopy (OM) and electron backscatter diffraction (EBSD). The results show that dynamic reversion (DRV) and dynamic recrystallization (DRX) are the main mechanisms of microstructure evolution during the thermal deformation of 7A85 aluminum alloy. 350–400 °C is the transition zone from dynamic reversion to dynamic recrystallization. DRV is the main mechanism at 350 °C, while DRX occurs mainly at 400 °C when the sensitivity of microstructural evolution to temperature is relatively high. Li et al. [22] investigated the effect of dynamic recrystallization (DRX) on the organization and mechanical properties of 6063 aluminum alloy extrusion profiles by experiments and simulations. The results show that increasing the stamping speed at a low stamping speed leads to an increase in DRX fraction due to the increase in temperature and strain rate. At high stamping speeds, further increases in stamping speed have a much smaller effect on temperature, and the DRX effect is reduced due to the high strain rate.

Therefore, it is important to study the dynamic recrystallization behavior of 6082-T6 aluminum alloy and the dynamic evolution of microstructure during thermal creep to optimize the heat deformation parameters and regulate the heat deformation organization and properties. In this paper, a gleeble-3500 thermal simulation compressor, metallurgical microscope (MM) and scanning electron microscope (SEM) are used to investigate the dynamic crystallization and creep behavior of 6082-T6 aluminum alloy based on rheological stress–strain data to represent DRX characteristics. Arrhenius instantonal equation, critical condition model and volume fraction model of DRX are developed. To provide guidance for the thermal processing process of this alloy.

## 2. Methods

The size of the experimental specimen is Φ8 mm × 12 mm. The specimens were heated to 613.15, 663.15, 713.15, and 763.15 K at 18 K/S by a gleeble-3500 thermal simulation compressor and then maintained for 3 min to make the temperature uniformly distributed, after which the specimens were thermally compressed at strain rates of 0.01, 0.1, 1, and 10 S^−1^, respectively, and the specimens were water-cooled immediately after the experiments to retain deformation. After all experiments are completed, the desired specimen is cut along the axial direction. After all the samples were polished, the cross sections of all the samples were analyzed by scanning electron microscope (SEM). The elemental composition of 6882 aluminum alloy is shown in Table 1.

## 3. Results

### 3.1. Stress–Strain Curve

Figure 1 shows the real stress–strain curve of hot compression creep of 6082-T6 aluminum alloy. From the curve changes in the figure, the hot compression process can be divided into three stages: work hardening, high-temperature softening, and dynamic equilibrium stabilization [23]. At the beginning of creep, the strain increases slowly, and the stress rises substantially because of the hardening effect that occurs. As the strain increases, dynamic reversion (DRV) and dynamic softening recrystallization (DRX) accompany the material internally, weakening the effect of work-hardening, and the stress rise tends to level off.

As creep proceeds, when the strain reaches a certain value, the stress–strain produces a dynamic equilibrium with a relatively horizontal curve and a distinct dynamic equilibrium [24,25]. As the increase in temperature causes active atomic motion within the material, it is prone to cross-slip and climbing of dislocations at high temperatures, which can lead to stress reduction [26]. Therefore, when the strain rate is constant, an increase in temperature leads to a decrease in stress; as the strain rate becomes larger, it directly affects the dislocations of atoms inside the material, and it will be accompanied by atomic entanglement and crossover phenomena, leading to a more difficult creep.

### 3.2. Micro-Analysis

Scanning electron microscope (SEM) was used to carry out microscopic analysis on four samples with test conditions of 663.15 K, 0.001 S^−1^, 663.15 K, 10 S^−1^, 713.15 K, 1 S^−1^, 763.15 K, 1 S^−1^, as shown in Figure 2, in which the figure with the lower corner mark in the figure number is the enlarged microscopic picture of the cross-section structure. Through observation, it is found that bright white, gray and black DRX crystals appear in the cross sections of these four samples, and all the crystals have different shapes. As can be seen from Table 1, among all the elements that make up 6082, aluminum alloy, silicon, iron, manganese and magnesium account for a large proportion compared with other elements. According to the physical characteristics of various elements, it is preliminarily inferred that the main phase of bright white DRX crystals is the Al-Mg phase and gray DRX crystals. In order to further understand the composition elements of the DRX crystal block, the SEM micrographs of the cross sections of these four samples were scanned, and the energy spectra of Mg, Al, Si and Mn elements were analyzed. The analysis results are shown in Figure 3.

Comparing Figure 3 with Figure 2, it is found that the black DRX crystal block in Figure 3a,b mainly contains the Mg-Si phase, and the white crystal block mainly contains the Mn phase; In Figure 3c,d, the black DRX crystal block mainly contains Mg-Si phase, while the white crystal block mainly contains Mn phase and Mn-Si phase; Through the analysis of Figure 3, the phase composition of the gray DRX crystal block can’t be judged. Therefore, the crystal block points marked in Figure 3 are scanned one by one according to the order of the marked points, and the final proportion of elements and atoms to each point is shown in Table 2. According to the analysis of Table 2, when the content of Mg and Si in the DRX crystal block is far greater than that of Mn, the crystal block will appear black; When the content of Mn and Si in the DRX crystal block is close, and the content of Mn and Si is far greater than that of MG, the crystal block will appear bright white. When the content of Mn and Si in the DRX crystal block is far greater than that of MG, and the content of Si is greater than that of Mn, the crystal block will appear bright white.

### 3.3. Thermal Creep Model

The Arrhenius hyperbolic sinusoidal instanton model [27,28] was used experimentally in the paper to analyze the effects of strain, strain rate and temperature on the thermal creep of 6082-T6 aluminum alloy with the following model expressions.
(1)$$\dot{\varepsilon }=A{\left[sinh\left(\alpha \sigma \right)\right]}^{n}exp\left(\frac{-Q}{RT}\right)$$

In the above equation, *Q* is the material heat deformation activation energy (J/mol); $\dot{\varepsilon }$ is the strain rate (s^−1^); *T* is the thermodynamic temperature (K); *R* is the gas constant (8.314 J/mol∙K); and *A*, *a*, and *n* are the material constants.

Different power function models are used to describe $\dot{\varepsilon }$ versus *σ* for different stress levels.
(2)$$\dot{\varepsilon }=A_{1}\sigma^{n_{1}}\left(\alpha \sigma >1.2\right)$$
(3)$$\dot{\varepsilon }=A_{2}exp\left(\beta \sigma \right)\;(\alpha \sigma <0.8)$$

In the above equation, $n_{1},\;\;A_{2},\;\;\beta ,\;\;A_{1}$ are material fixed properties.

Taking the natural logarithm of both sides of Equations (1)–(3) to collate gives.
(4)$$ln\left(\dot{\varepsilon }\right)=lnA+nln\left[sinh\left(\alpha \sigma \right)\right]-Q/RT$$
(5)$$ln\left(\dot{\varepsilon }\right)=lnA_{1}+n_{1}ln\sigma$$
(6)$$ln\left(\dot{\varepsilon }\right)=lnA_{2}+\beta \sigma$$

The relationship between $ln\left(\dot{\varepsilon }\right)-\sigma$ and $ln\left(\dot{\varepsilon }\right)-ln\sigma$ is shown in Figure 4a,b, which can be fitted with β=0.217035, $n_{1}=9.98189$.
(7)$$\alpha =\beta /n_{1}$$

From Equation (7), we can find α=0.02174. Bringing the experimental environment as strain rate variation, strain rate, peak stress and α under constant temperature conditions into Equation (4), the image of $ln\left(\dot{\varepsilon }\right)-ln\left[sinh\left(\alpha \sigma \right)\right]$ is plotted as shown in Figure 4c and *n* = 6.77587 is obtained.

Magnifying 1/T in Equation (1) by a factor of 1000 and taking the logarithm of both sides of the equal sign, the collation gives.
(8)$$ln\left[sinh\left(\alpha \sigma \right)\right]=\frac{ln(\dot{\varepsilon })}{n}-\frac{lnA}{n}+\frac{1000Q}{nRT}$$

Bringing the experimental data into Equation (8), the relation ln[sinh(ασ)]−1/T is plotted as shown in Figure 4d, and its partial differentiation yields.
(9)$$Q=R{\left[\frac{\partial ln\dot{\varepsilon }}{\partial ln\left[sinh\left(\alpha \sigma \right)\right]}\right]}_{T}{\left[\frac{\partial ln\left[sinh\left(\alpha \sigma \right)\right]}{\partial \left(1/T\right)}\right]}_{\dot{\varepsilon }}$$

The three terms to the right of the middle sign of the above equation are R (gas constant), the slope of the function $ln\dot{\varepsilon }-ln[sinh\left(\alpha \sigma \right)$ at the same temperature conditions and the slope of the function ln[sinh(ασ)−1000/T at the same strain rate. At this moment, all parameters are obtained, and the average deformation activation energy Q=254.473kJ/mol of 6082-T6 aluminum alloy is obtained by bringing in the relevant parameters.

To couple the temperature and strain rate on creep behavior, the Zener-Holloman parameter Z [29] was introduced to describe the relationship between the three.
(10)$$Z=\dot{\varepsilon }exp(Q/RT)=A{\left[sinh\left(\alpha \sigma \right)\right]}^{n}$$

Bringing in the relevant parameters obtained in the previous section, we obtain.
(11)$$Z=\dot{\varepsilon }exp\left(257473/RT\right)$$

Taking the logarithm of Equation (10) yields.
(12)lnZ=lnA+nln[sinh(ασ)]

Combining the previously calculated values of ln[sinh(ασ) and deformation activation energy Q and other influencing parameter values (10) in Equation (1), the relationship of lnZ−ln[sinh(ασ)] is plotted as shown in Figure 5, and the fitted parameter $A=5.13411\times 10^{18}$ is obtained, and the obtained parameters are brought into Equation (1) to determine the 6082-T6 aluminum alloy in the temperature variation range of 340~490 °C and the strain rate variation range of 0.01~10 S^−1^, the intrinsic model of thermal creep of 6082-T6 aluminum alloy is determined.
(13)$$\dot{\varepsilon }=5.13411\times 10^{18}{\left[sinh\left(0.02174\sigma \right)\right]}^{6.77587}\times exp\left(-257473/RT\right)$$

### 3.4. Creep Crystallization Critical Conditions

The key to DXR and thermal processing optimization technology is to study the critical conditions for DXR. The work hardening rate (θ=dσ/dε) is introduced to reflect the trend of material rheological stress with strain, and the work hardening rate-stress (θ−σ) curve is plotted to determine the critical stress under different conditions. In studying the critical stress of thermal creep of materials, Najafi Zadeh et al. [30,31] proposed that (θ−σ) images can be well fitted to the cubic polynomial, and the critical condition of DRX can be determined if the inflection point is determined based on (dθ/dσ) – σ images [32]. The DRX critical condition determination criterion can be determined in the critical model based on the P-J criterion [33,34].

Figure 6 shows the corresponding cubic polynomial fits of the θ – σ curves for specimens at different temperatures from 0.01 to 10 S^−1^. It can be seen from the figure that each curve intersects the horizontal axis and has two intersection points, the first intersection point is the softening stress ($\sigma_{s}$) and the second intersection point is the peak stress ($\sigma_{p}$). To determine the curve inflection point and judge the critical condition, the (−dθ/dσ) – σ curve needs to be plotted. Figure 7 shows the (−dθ/dσ) – σ curves for different creep conditions. The stress value corresponding to the lowest point of this curve is critical stress.

From the change of curves in Figure 7, it can be seen that the increase in temperature and the decrease in strain rate will lead to the subsequent decrease of critical stress; the increase of temperature will enhance the rate of atomic motion inside the material, which is prone to interatomic dislocation slip and promote DRX nucleation, thus decreasing the creep dynamic recrystallization critical condition; the decrease of strain rate is less likely to lead to a large number of dislocation entanglement, and the stress generated by dislocation entanglement is more dispersed, thus increasing the DRX nucleation rate and leading to the decrease of critical condition.

Figure 8 shows the relationship between $\varepsilon_{c}-\varepsilon_{p}$ and $\sigma_{c}-\sigma_{p}$ for all specimens, and Figure 9 shows the relationship between critical stress $\sigma_{c}$ (critical strain $\varepsilon_{c}$), peak stress $\sigma_{p}$ (peak strain $\varepsilon_{p}$) and parameter Z. It can be seen from the figure that the relationship between the parameters is linear, and the following relationship is obtained by linear fitting of the data.
(14)$$\{\begin{array}{c} \varepsilon_{c}=0.86614\varepsilon_{p}\\ \sigma_{c}=0.93002\sigma_{p}\\ \varepsilon_{p}=0.003345exp\left(0.9209lnZ\right)\\ \sigma_{p}=0.424077exp\left(0.10543lnZ\right)\\ \varepsilon_{c}=0.001636exp\left(0.10089lnZ\right)\\ \sigma_{c}=0.218286exp\left(0.11785lnZ\right) \end{array}$$

According to Figure 9 and Equation (14), the increase in critical and peak stress–strain is positively related to the increase in parameter Z (positively related to strain rate and negatively related to temperature). More recrystallized grains can be observed in Figure 2b because there is enough energy and time for crystallization and grain boundary migration to occur at low Z values, making DRX more likely to occur. As can be seen from Figure 3b, a large number of dislocations will accumulate at high Z values, leading to an increase in the interaction between dislocations and grain boundaries, making the stress concentration at grain boundaries unfavorable to the occurrence of dynamic recrystallization and weakening the softening effect.

### 3.5. Creep Recrystallization Volume Model

Modeling creep recrystallization volume fraction is critical for improving the performance of 6082 aluminum alloy materials and for studying the DRX behavior during thermal creep [35]. In this paper, the Avrami model is used to construct a dynamic recrystallization kinetic model [36]. The model expression is as follows.
(15)$$X=1-exp\left\{-k{\left[\left(\varepsilon -\varepsilon_{c}\right)/\varepsilon_{p}\right]}^{m}\right\}$$

The *k* and *m* in the above equation are inherent material properties. In this paper, the stress softening model is used to find the value of *X* under different conditions [37] in order to construct the Av rami model.
(16)$$X=\frac{\sigma_{sat}-\sigma }{\sigma_{sat}-\sigma_{s}}$$

In the above equation, $\sigma_{sat}$ is the dynamic reversion stress, which can be replaced by the peak stress $\sigma_{p}$ in the calculation, and $\sigma_{s}$ is the softening stress in the steady state section after the creep recrystallization process [38].

Taking the natural logarithm of both sides of the equal sign of Equation (15) yields
(17)$$lnln\;\left[1/\left(1-X\right)]=mln[\left(\varepsilon -\varepsilon_{c}\right)/\varepsilon_{p}\right]+lnk$$

Figure 10 shows the graph of $\mathrm{lnln}\;\left[1/\left(1-X\right)]-\ln[\left(\varepsilon -\varepsilon_{c}\right)/\varepsilon_{p}\right]$ fitted relationship for strain rate at 0.01 and 10 S^−1^. The corresponding m = 2.75042 and *k* = 0.101692 at a strain rate of 0.01 S^−1^ and *m* = 2.46052 and *k* = 0.008952 at a strain rate of 10 S^−1^ are obtained by fitting. Creep recrystallization equations for the corresponding conditions can be obtained by bringing in the relevant parameters.
(18)$$\{\begin{array}{c} X=1-exp\left\{-0.101692{\left[\left(\varepsilon -\varepsilon_{c}\right)/\varepsilon_{p}\right]}^{2.75042}\right\}\\ X=1-exp\left\{-0.008952{\left[\left(\varepsilon -\varepsilon_{c}\right)/\varepsilon_{p}\right]}^{2.46052}\right\} \end{array}$$

The predicted creep recrystallization volume fraction curves for 6082-T6 aluminum alloy at strain rates of 0.01 and 10 S^−1^ are shown in Figure 11, based on the kinetic equations obtained in the previous section. As can be seen from Figure 11, the DRX volume fraction of the material rises very slowly at the beginning of creep; along with the increase in strain, the DRX volume fraction rises rapidly, which is consistent with the variation law of the DRX volume fraction of aluminum alloys by S Ding et al. [39]. A positive relationship between the increase in DRX volume fraction and the decrease in strain rate at a constant temperature.

When the strain rate is constant, the increase in DRX volume fraction is proportional to the increase in temperature; the increase in temperature will give the recrystallization nucleation energy and promote the nucleation to become larger, resulting in the increase of DRX volume fraction. The increase in strain rate will generate a large number of atomic dislocations, resulting in a weakening of the ability to nucleate at the crystallization site, resulting in a decrease in DRX volume fraction. By comparing the metallographic diagrams of the specimens, the recrystallization equations obtained in this paper are in good agreement with the experimental phenomena, indicating that the DRX model of 6082-T6 aluminum alloy can reflect the variation of DRX volume fraction more accurately. Therefore, the creep conditions can be changed to improve the microstructure of the material to achieve the purpose of improving the material properties.

## 4. Conclusions

The value of thermal creep activation energy Q of 6082-T6 aluminum alloy is 254.473 kJ/mol, and the constructed thermal creep principal equation is:


(19)
$$\dot{\varepsilon }=5.13411\times 10^{18}{\left[sinh\left(0.02174\sigma \right)\right]}^{6.77587}\times exp\left(-257473/RT\right)$$


2.Based on the Avrami model, the kinetic equations for creep crystallization of 6082-T6 aluminum alloy at 613.15 to 763.15 K with strain rates of 0.01 and 10 S^−1^ are constructed as follows.


$$X=1-exp\left\{-0.101692{\left[\left(\varepsilon -\varepsilon_{c}\right)/\varepsilon_{p}\right]}^{2.75042}\right\}$$



$$X=1-exp\left\{-0.008952{\left[\left(\varepsilon -\varepsilon_{c}\right)/\varepsilon_{p}\right]}^{2.46052}\right\}$$


3.Scanning electron microscope analysis shows that DRX crystal blocks have different colors and shapes. When the contents of Mg and Si elements in DRX crystal blocks are far greater than that of Mn elements, the crystal blocks will appear black. When the content of Mn and Si in the DRX crystal block is close, and the content of Mn and Si is far greater than that of MG, the crystal block will appear bright white. When the content of Mn and Si in the DRX crystal block is far greater than that of MG, and the content of Si is greater than that of Mn, the crystal block will appear bright white.4.The critical conditions for creep crystallization of 6082-T6 aluminum alloy were described by introducing the work-hardening rate (θ) and parameter Z. The critical conditions for this material were obtained as follows: $\varepsilon_{c}=0.86614\varepsilon_{p}$; $\sigma_{c}=0.93002\sigma_{p}$; $\varepsilon_{p}=0.003345exp\left(0.9209lnZ\right)$; $\sigma_{p}=0.424077exp\left(0.10543lnZ\right)$; $\varepsilon_{c}=0.001636exp\left(0.10089lnZ\right)$. $\sigma_{c}=0.218286exp\left(0.11785lnZ\right)$; the parameter $Z=\dot{\varepsilon }\exp\left(257473/\mathrm{RT}\right)$ in the previous equation, it is found that the critical condition is positively correlated with the increase of parameter Z, positively correlated with the increase of strain rate, and negatively correlated with the increase of temperature.

## Figures and Tables

**Figure 1 materials-15-08117-f001:**
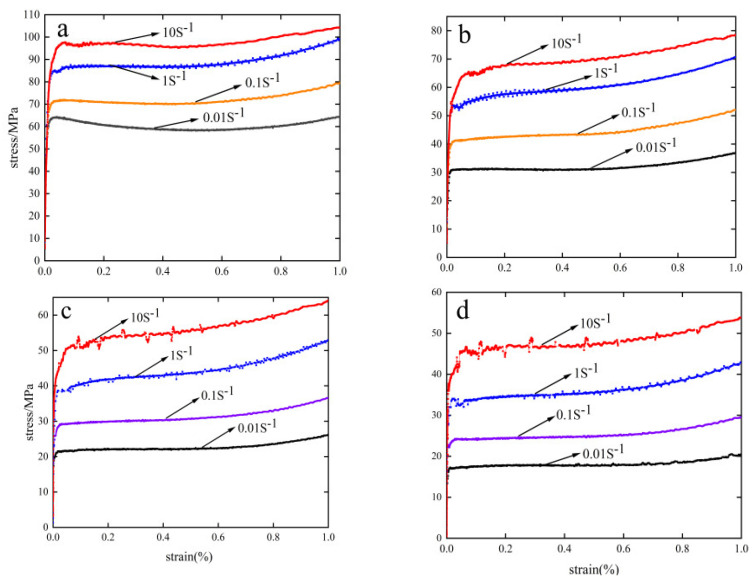
Real stress–strain curves: (**a**) 613.15 K, (**b**) 663.15 K, (**c**) 713.15 K, (**d**) 763.15 K.

**Figure 2 materials-15-08117-f002:**
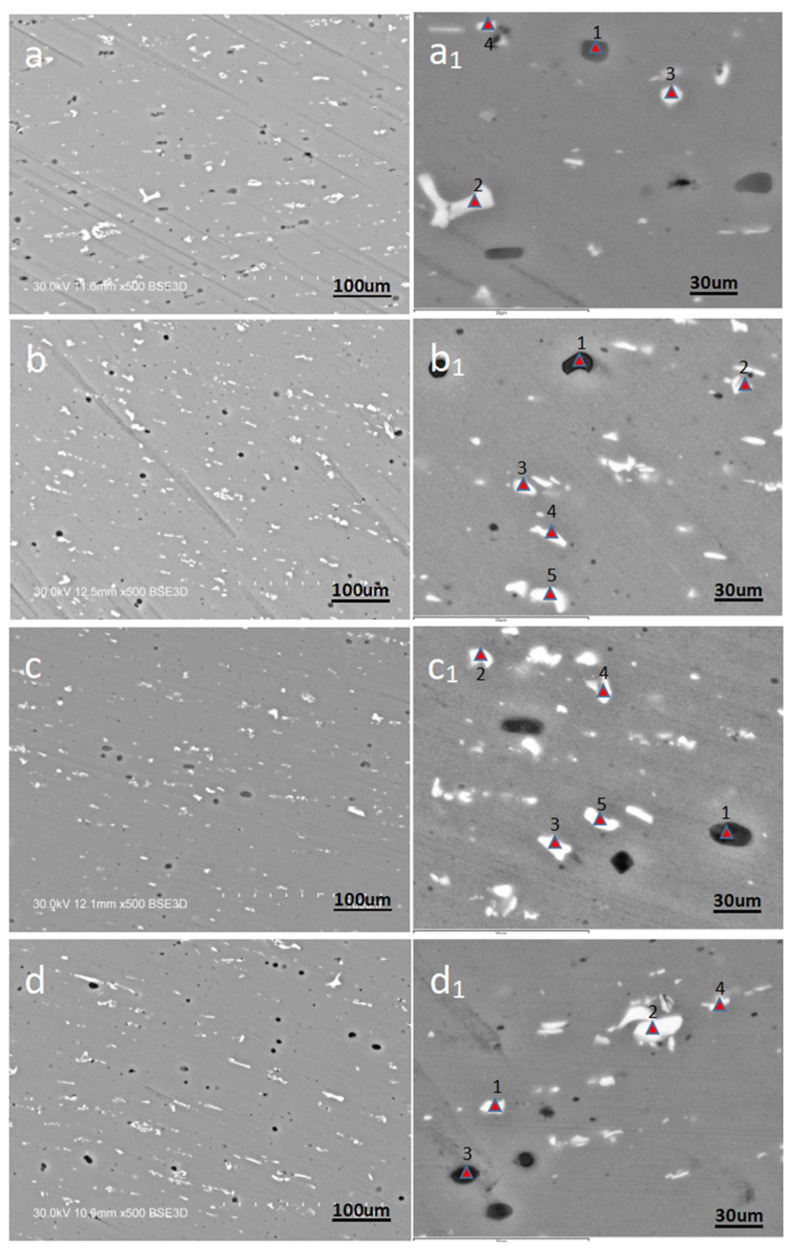
SEM microstructure of some samples ((**a**,**a_1_**): 663.15 K, 0.001 S^−1^ (**b**,**b_1_**): 663.15 K, 10 S^−1^ (**c**,**c_1_**): 713.15 K, 1 S^−1^ (**d**,**d_1_**): 763.15 K, 1 S^−1^).

**Figure 3 materials-15-08117-f003:**
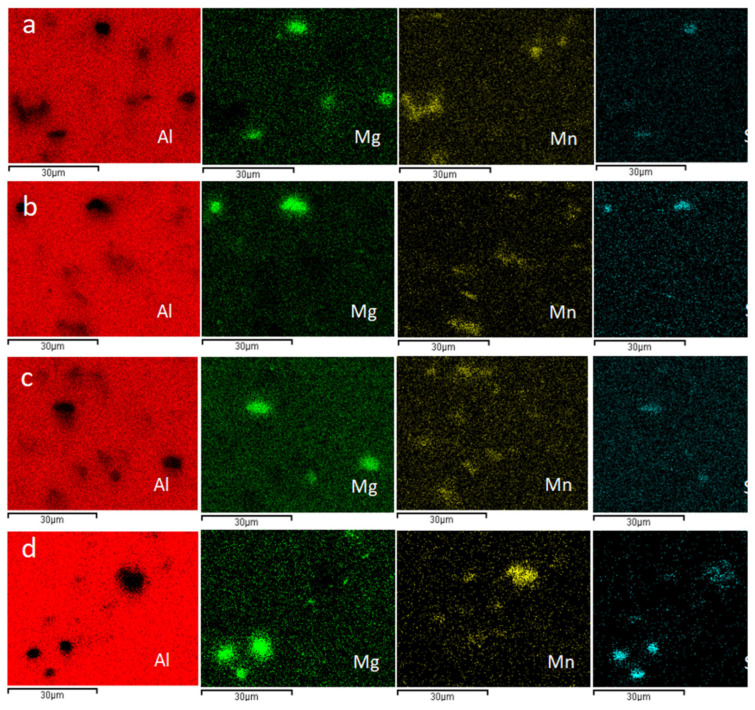
Element surface scanning (**a**) 663.15 K, 0.001 S^−1^, (**b**) 663.15 K, 10 S^−1^, (**c**) 713.15 K, 1 S^−1^, (**d**) 763.15 K, 1 S^−1^.

**Figure 4 materials-15-08117-f004:**
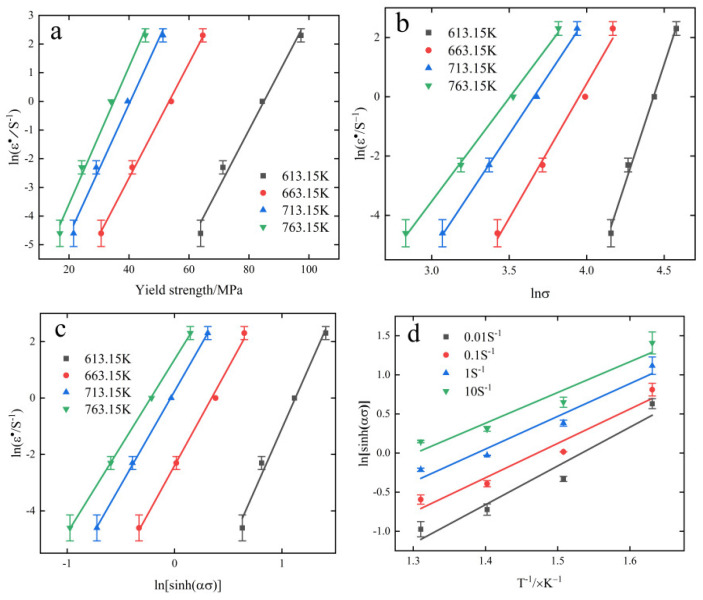
(**a**) $ln\left(\dot{\varepsilon }\right)-\sigma$, (**b**) $ln\left(\dot{\varepsilon }\right)-ln\sigma$, (**c**) $ln\left(\dot{\varepsilon }\right)-ln\left[sinh\left(\alpha \sigma \right)\right]$ and (**d**) $ln\left[sinh\left(\alpha \sigma \right)\right]-\frac{1}{T}$.

**Figure 5 materials-15-08117-f005:**
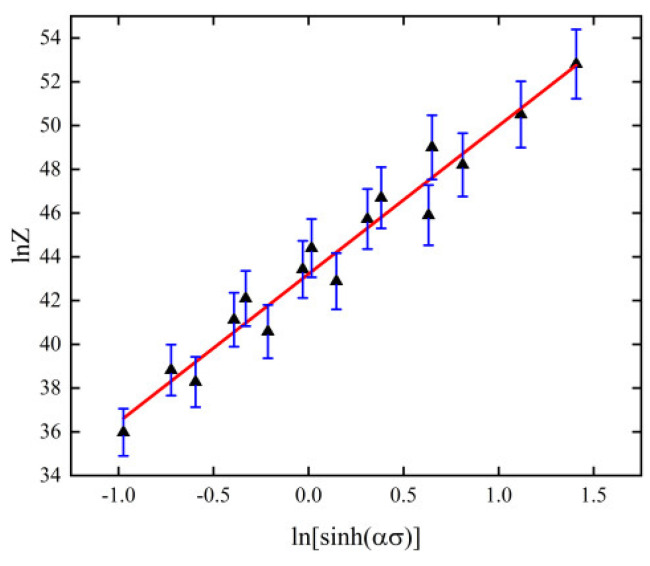
lnZ−ln[sinh(ασ)] relationship curve..

**Figure 6 materials-15-08117-f006:**
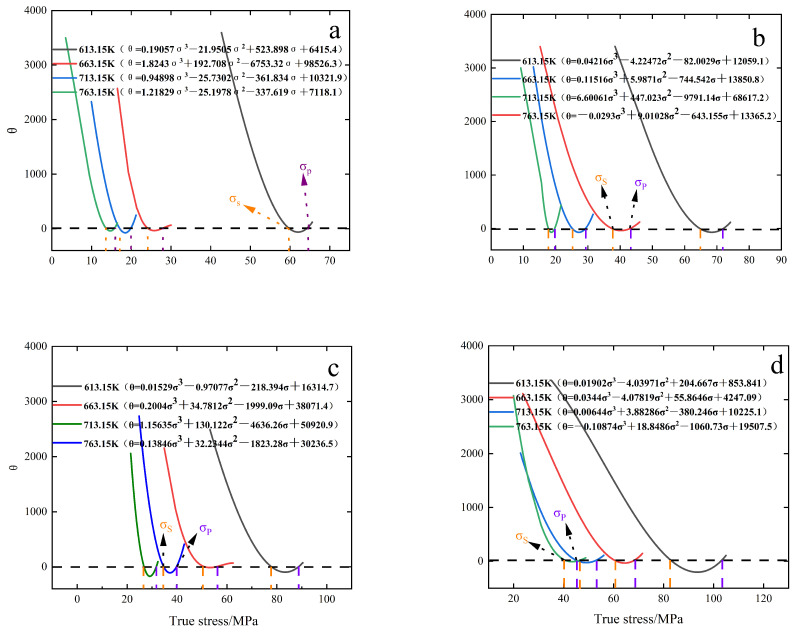
θ−σ curves at different strain rates. (**a**) 0.01 S^−1^.; (**b**) 0.1 S^−1^; (**c**) 1 S^−1^; (**d**) 10 S^−1^.

**Figure 7 materials-15-08117-f007:**
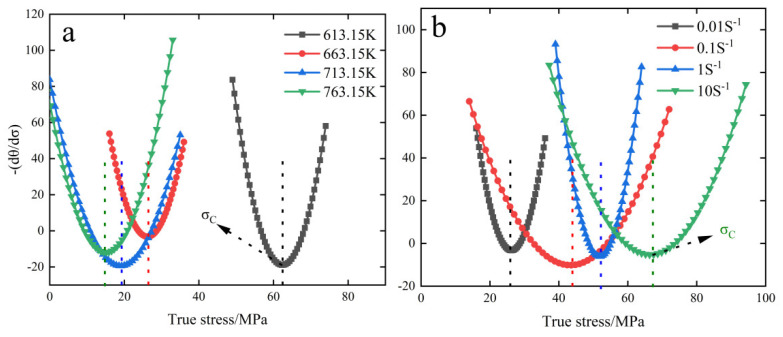
(−dθ/dσ) − σ curves for different creep conditions: (**a**) 0.01 S^−1^, (**b**) 663.15 K.

**Figure 8 materials-15-08117-f008:**
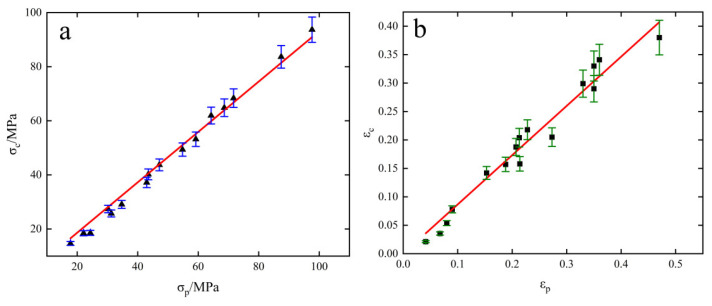
Critical condition fitting relationship. (**a**)$\;\sigma_{c}-\sigma_{p}$; (**b**)$\;\varepsilon_{c}-\varepsilon_{p}$.

**Figure 9 materials-15-08117-f009:**
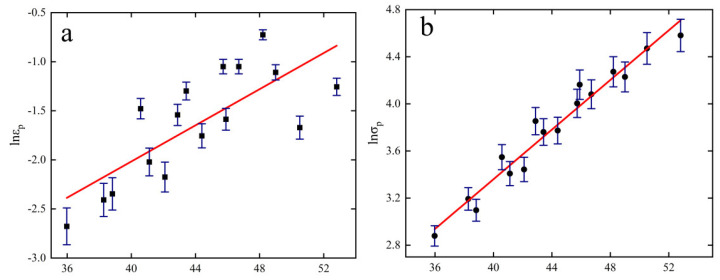
Fit between critical conditions and Z. (**a**) $\;ln\varepsilon_{p}-lnZ$; (**b**) $\;ln\sigma_{p}-lnZ$; (**c**) $\;ln\varepsilon_{c}-lnZ$; (**d**) $ln\sigma_{c}-lnZ$.

**Figure 10 materials-15-08117-f010:**
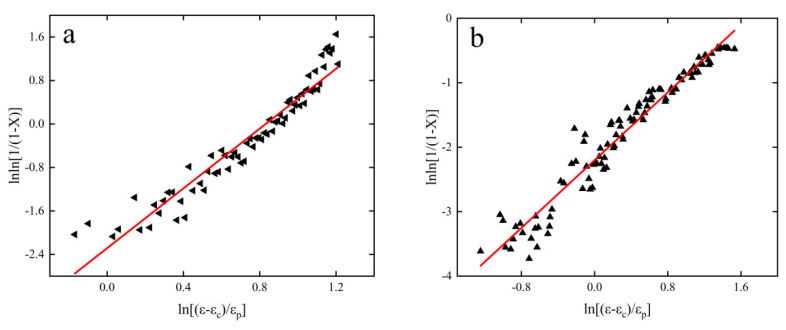
$lnln\;\left[1/\left(1-X\right)]-ln[\left(\varepsilon -\varepsilon_{c}\right)/\varepsilon_{p}\right]$ fit plots: (**a**) 0.01 S^−1^, (**b**) 10 S^−1^.

**Figure 11 materials-15-08117-f011:**
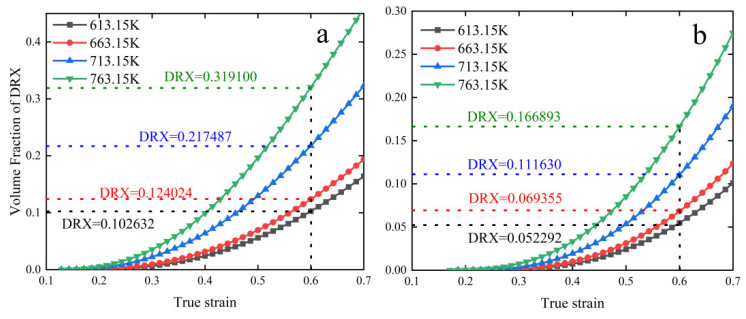
Crystallization volume fraction prediction curves: (**a**) 0.01 S^−1^, (**b**) 10 S^−1^.

**Table 1 materials-15-08117-t001:** 6082 Element Composition of Aluminum Alloy.

Si	Fe	Cu	Mn	Mg	Cr	Zn	Ti	Al
0.7~1.3	0.50	0.10	0.4~1.0	0.6~1.2	0.25	0.20	0.10	Bal

**Table 2 materials-15-08117-t002:** Elements and atomic contents of each point (%).

	Element		Mg	Al	Si	Mn		Mg	Al	Si	Mn
a_1_	weight	Point 1	18.37	58.54	22.93	0.16	Point 2	0.07	80.73	9.57	9.63
	atom		20.18	57.94	21.81	0.08		0.08	85.22	9.71	4.99
	weight	Point 3	0.33	80.76	9.16	9.75	Point 4	0.32	87.64	7.46	4.58
	atom		0.39	85.27	9.29	5.05		0.37	89.97	7.36	2.31
b_1_	weight	Point 1	15.65	67.51	16.53	0.31	Point 2	0.21	86.73	8.15	4.92
	atom		17.22	66.90	15.73	0.15		0.24	89.22	8.05	2.48
	weight	Point 3	0.21	87.96	7.17	4.66	Point 4	0.21	86.06	8.28	5.45
	atom		0.24	90.33	7.08	2.35		0.24	88.79	8.20	2.76
	weight	Point 5	0.39	85.30	8.36	5.94					
	atom		0.45	88.22	8.31	3.02					
c_1_	weight	Point 1	13.25	67.18	19.32	0.24	Point 2	0.20	85.54	7.90	6.35
	atom		14.62	66.80	18.45	0.12		0.23	88.66	7.87	3.23
	weight	Point 3	0.41	88.19	7.47	3.93	Point 4	0.28	86.98	8.09	4.65
	atom		0.47	90.21	7.34	1.98		0.32	89.35	7.99	2.34
	weight	Point 5	0.28	86.27	8.33	5.13					
	atom		0.32	88.85	8.24	2.59					
d_1_	weight	Point 1	0.62	89.04	6.27	4.07	Point 2	0.21	75.53	11.23	13.04
	atom		0.71	91.09	6.16	2.04		0.25	81.26	11.60	6.89
	weight	Point 3	11.40	68.18	20.15	0.27	Point 4	0.40	89.26	6.72	3.62
	atom		12.61	67.96	19.30	0.13		0.46	91.14	6.59	1.82

## Data Availability

Not applicable.

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
