# Peer review of "Thermal Creep Behavior and Creep Crystallization of Al-Mg-Si Aluminum Alloys"

_materials, 2022, doi:10.3390/ma15228117_

Round 1

Reviewer 1 Report

Dear Authors,

Please find below my comments/observations regarding your manuscript:

1. For the “Methods” part: Please add motivations regarding the selection of the used processing parameters. Why were the five processing/heating temperatures and the five strain rates specifically chosen?

2. For the “Methods” part: Please explain the following sentence: “Due to the weak conductivity of the aluminum alloy material, the surface of the SEM specimen needs to be coated with gold spray to enhance the conductivity”. Why the SEM specimen needs to be coated with gold, knowing that the thermal conductivity of aluminum and Al-alloys is not weak, but is about three times greater than that of steel?

3. Please revise the Figure 1. It is totally unclear. The same for Figures 4-11

4. For the Figure 1 please add SI unit [%] for the strain.

5. It follows from the text and Figure 2 that the deformation conditions for the two variants from Fig.2-b and 2-c are identical (713.15K/0.1S-1).  Please revise.

6. First paragraph from #3.2 Micro-analysis # contains comments that cannot be covered by figure 2; these kinds of microstructural processes cannot be visible at the resolution of the image from figure 2 (dislocation movements, atomic mobility, etc.). that's why, from my point of view, figure 2 is useless, it can be cut from the text; moreover, it doesn't even show notable differences between the four variants, because the magnification is too small. Instead, it can be replaced with another one from a SEM or TEM examination, with a much higher magnification. Even so, SEM images can show dislocations, twins, etc., only for very high magnification.

7. It is not clear: The text indicates that both fig 2 and 3 represent “microstructure photographs of the specimens under different creep conditions” but fig 2 indicate a planar surface and fig 3 indicate a breaking surface. This difference must be clarified in the text. Then, the text specified: “TEM photographs at 613.15 K/0.1 S-1 and 763.15 131 K/0.1 S-1 are shown in Figure 4c and Figure 4d”, but images of TEM microscopy doesn’t exist in the manuscript.

8. The comments corresponding to Fig 3 have also no coverage with the provided images. No “dislocation entanglements around the grains” are seen. It is only a breaking surface. From the same reasons, the following statement is also incorrect: “From Figure 3c, it can be seen that a small amount of DRX grains and entanglement dislocations exist at the grain boundaries”. All notes and circles with red color in figure 3 are incorrectly indicated, they cannot signal the mentioned microstructural details.

9. Due to comments 3-8 above, I suggest to revise substantially the part corresponding to microstructure analysis.   

10. The caption of the Figure 4 must be more clearly/detail explained.

11. The conclusion no.3 represents assertions already considered classic in physical metallurgy, demonstrated and experienced countless times; so, nothing original. The calculations made on the particular case of the studied alloy can be considered as a calculation exercise that can be replicated for other alloys as well.

Author Response

See the annex for the reply to the modification comments.

Reviewer 2 Report

* Change the word "Av rami" for "Avrami" along the paper

* The meaning of acronims DRV and DXR should be described at line 35 and not at line 49

* Line 38: change "L et al" to "Zhen et al"

* The authors should indicate the meaning of AF, BF, MF and EF at line 43

* No differences can be seen between figures 2a to 2d. Those differences are described in lines 103 to 119, but image 2 should also show those differences, perhaps using a higher magnification.

* Lines 131-132: The authors cite figures 4c and 4d as TEM photographs, but those figures do not exist.

* Figure 3: The authors should indicate how the samples for SEM observation was obtained, as they where not obtained by cutting and polishing. Do they correspond to the surface of a crack?

* Regarding figure 3, the authors talk about dislocations as if they were visible using a so low magnification. They should clarify this point.

* Line 152: Change "a" to the greek letter "alpha"

* Why do the authors need different models to study  the results? (equations 1-3) 

* lines 170, 171 and equation 9: A bracket "]" is missing in some equations.

* I don't understand why the authors need to convert all data to fit a linear equation. There are enough mathematical fitting methods to not need this technique. The paper could be reduced in size significantly.

* Line 185: "to determine...". I think something is missing

Author Response

(The authors gave the same response as above.)

Round 2

Reviewer 1 Report

No anymore comments to make. Thank you very much.